# Identification and Validation of Key Genes Related to Preferred Flavour Profiles in Australian Commercial Papaya (*Carica papaya* L.)

**DOI:** 10.3390/ijms25053046

**Published:** 2024-03-06

**Authors:** Ziwei Zhou, Chutchamas Kanchana-udomkan, Rebecca Ford, Ido Bar

**Affiliations:** 1Centre for Planetary Health and Food Security, School of Environment and Science, Griffith University, Nathan, QLD 4111, Australia; ziwei.zhou@griffith.edu.au (Z.Z.); c.kanchana-udomkan@griffith.edu.au (C.K.-u.); i.bar@griffith.edu.au (I.B.); 2Department of Horticulture, Faculty of Agriculture at Kamphaeng Saen, Kasetsart University Kamphaeng Saen Campus, Nakhon Pathom 73140, Thailand

**Keywords:** RNA sequencing, sweetness genes, RT-qPCR, differential expression, de novo genome assembly

## Abstract

Commercial papaya varieties grown in Australia vary greatly in taste and aroma. Previous profiling has identified undesirable ‘off tastes’ in existing varieties, discouraging a portion of the population from consuming papayas. Our focus on enhancing preferred flavours led to an exploration of the genetic mechanisms and biosynthesis pathways that underlie these desired taste profiles. To identify genes associated with consumer-preferred flavours, we conducted whole RNA sequencing and de novo genome assembly on papaya varieties RB1 (known for its sweet flavour and floral aroma) and 1B (less favoured due to its bitter taste and musty aroma) at both ripe and unripe stages. In total, 180,368 transcripts were generated, and 118 transcripts related to flavours were differentially expressed between the two varieties at the ripe stage. Five genes (*cpBGH3B*, *cpPFP*, *cpSUS*, *cpGES* and *cpLIS*) were validated through qPCR and significantly differentially expressed. These genes are suggested to play key roles in sucrose metabolism and aromatic compound production pathways, holding promise for future selective breeding strategies. Further exploration will involve assessing their potential across broader germplasm and various growth environments.

## 1. Introduction

Papaya (*Carica papaya* L.) is an important tropical fruit crop cultivated worldwide. Global production has steadily increased over the past twenty years, reaching 13.7 million tonnes in 2019 [1]. The Australian papaya industry is relatively small, with only 18.3 thousand tonnes grown annually [2]. One of the major challenges to the Australian papaya industry is ensuring that fruit have consumer-preferred flavours. The commercial papaya varieties grown in Australia vary greatly in their flavours, ranging from unpleasant aromas or bitter tastes to preferred aromas and sweeter fruit [3]. To improve the reliability of the fruit flavour and increase consumer acceptance of Australian papaya, it is necessary to better understand the genetic mechanisms and biosynthesis pathways of candidate genes underpinning the desirable flavour traits. This, in turn, will support future selective breeding programs to develop better-tasting new commercial papaya varieties.

Sugar and volatile compounds are major factors contributing to papaya flavour and aroma. Flesh sweetness is determined by three main soluble sugars, which are glucose, fructose and sucrose [4]. The synthesis of these sugars throughout the ripening stage in papaya is mediated by the key enzymes invertase, sucrose phosphate synthase (SPS) and sucrose synthase (SS) [5,6]. Volatiles already known to contribute to papaya aroma at the ripening stage are methyl and ethyl ester derivatives of lipid catabolism [7]. A more recent study of flavour profiling of Australian commercial papaya varieties indicated that glucose, linalool oxide and terpinolene are the three most important biochemical compounds that influence consumer preference [3]. Thus, it is essential to identify and validate the functions of the key genes that encode enzymes and regulatory sequences in the biosynthetic pathways of these compounds in papaya.

RNA sequencing (RNA-Seq) is a technique used to investigate gene functions for traits of interest and has become a primary approach for transcriptome research due to its sensitivity in the detection of potentially important gene sequences that are expressed at low levels. However, the lack of a densely covered and fully annotated papaya reference genome is an immediate limitation to the use of RNA-Seq analysis in papaya. To date, the most comprehensive publicly available reference genome of papaya was published by Yue’s group [8], yielding 351.5 and 350.3 Mb, with 97.5% and 97.3% genome completeness, of the transgenic ‘SunUp’ and its progenitor ‘Sunset’ genomes, respectively [8]. The availability of a more dense and well-annotated genome assembly is critical for future advanced papaya genomics research. In cases where a reference genome or reference transcriptome is not available, de novo assembly (without the reference genome sequence) is an optional approach for transcriptome profiling and differential gene expression analysis [9]. Previously, de novo transcriptome assembly from RNA-Seq data has proven to be a useful tool to investigate gene expression of volatile compounds conferring peach-like aromas in strawberries (*Fragaria × ananassa*) [10]. Therefore, the aims of this study are to (1) identify putative sweetness- and volatile-related genes through whole RNA sequencing followed by (2) de novo assembly and differential expression analysis of two Australian commercial papaya varieties, *C. papaya* ‘RB1’ and *C. papaya* ‘1B’, which have distinct flavour profiles, and, together, these will be used to (3) discover the biosynthetic pathways underpinning preferred papaya flavours.

## 2. Results

### 2.1. RNA-Sequencing Assembly

A total number of 7.40 × 10^8^ paired-end raw reads with an average length of 100 bp were generated through Illumina NovaSeq (S2) from 12 RNA samples (RNA sample quality report was generated in Appendix A). After trimming adapters and removing low-quality bases, 3.67 × 10^8^ high-quality clean reads were produced with an average length of 87 bp. The mean alignment rate of the reads when mapped to the de novo transcriptome was 98.18%, which is higher than the value for the non-GMO ‘sunset’ genome (81.17%). The transcriptome of papaya was then de novo assembled to yield 180,368 transcripts (Table 1). These transcripts were then annotated to known genes against the nucleotide and protein database through NCBI while limiting the search to flowering plants, fungi, viruses and bacteria. The results indicated that around 137,855 (76.43%) of the transcripts were successfully matched and assigned with 26,089 accession numbers. Among them, 23,267 accession numbers matched *Carica papaya* sequences, 17,347 of them (74.55%) annotated to known genes, while the remaining 5920 accession numbers matched uncharacterised sequences whose functions are unknown. The transcript count data for each sample were used to produce a PCA (Figure 1). The spread of samples on the first and second largest eigenvalues indicated that the majority of variance in the count data was contributed by genotype variation (45%), while maturity stage (unripe or ripe stage) provided another 32% of the variability. The three replicates of each sampling group clustered well together, except one of the replicates of ‘RB1’ at the ripe stage, which was identified as being closer to the ‘unripe’ group. This was then treated as an outlier and removed from downstream expression analyses. The assembled sequences were deposited in the NCBI database (accession number GSE213436).

### 2.2. Differential Expression of Transcripts from RNA-Seq Data

In total, 27,545 transcripts were found to be differentially expressed between ‘RB1’ and ‘1B’ at the ripe stage based on the specified thresholds (‘FDR < 0.05’ and ‘|log2FC| > 1.5’). With ‘RB1’ set as the control group, 15,880 transcripts showed lower expressions in ‘1B’, and 11,665 transcripts were more highly expressed in ‘1B’ (Figure 2a). These differentially expressed transcripts were then assigned to functional pathways by their GO and KEGG terms.

Based on the KEGG enrichment analysis, DETs were matched to 2637 KO entries and assigned to 383 pathways. The most enriched were metabolic, biosynthesis of secondary metabolites, microbial metabolism in diverse environments and carbon metabolism pathways. There were significantly higher numbers of KOs involved in the metabolism of carbohydrates (166 KOs, 15 pathways), amino acids (123 KOs, 14 pathways) and lipids (102 KOs, 14 pathways) (Figure 2b). Moreover, in total, 12 pathway modules were completely generated from KEGG Mapper, 2 of which were closely related to sweetness and volatile compound production (modules M00854 and M00927). Module M00854 is involved in carbohydrate metabolism and functions in glycogen biosynthesis, converting glucose-1-phosphate to glycogen or starch. Module M00927 relates to plant terpenoid biosynthesis and is proposed to have a role in gibberellin A12 biosynthesis, converting geranylgeranyl pyrophosphate to gibberellin A12.

In total, 901 GO terms were successfully assigned from 7383 DETs with significant *p*-values (<0.05). Among these, 38 GO terms were assigned which were significantly different between ‘RB1’ and ‘1B’ at the ripe stage. These were then classified into three functional groups: biological processes (BP), cell cycle (CC) and molecular function (MF) (Figure 2c). Based on the enrichment results, the main GO terms enriched in DETs shared between ‘RB1’ and ‘1B’ at the ripe stage were associated with oxidoreductase activity from the MF classification as well as metal ion transport and fatty acid biosynthetic processes from the BP classification.

In this study, 118 DETs related to sugar and volatile metabolism pathways were identified (Appendix A). Among them, 86 were associated with carbohydrate metabolism, including glycolysis/gluconeogenesis and fructose and mannose metabolism, as well as starch and sucrose metabolism. Of these, 50 showed higher expression levels in ‘RB1’, while overall higher expression levels of DETs encoding beta-glucosidase were found in ‘1B’ compared to ‘RB1’. The remaining 32 were related to fatty acid and terpenoid biosynthesis, of which 17 were more highly expressed in ‘RB1’.

### 2.3. Validation of Functional Genes Related to Flavour Using RT-qPCR between ‘RB1’ and ‘1B’ Papaya

PCR products of the expected sizes were successfully amplified from each of the eight selected target genes (Figure 3). After primer efficiency testing, a consistent linear amplification was produced based on the R^2^ results and efficiency values. Differences among transcription levels of the selected genes were observed between the ‘RB1’ and ‘1B’ varieties when assessed at the ripe stage (Figure 4). These are proposed to encode for the following enzymes: beta-glucosidase (BGLU), pyrophosphate-fructose 6-phosphate 1-phosphotransferase (PFP) and sucrose synthase (SUS), involved in glycolysis/gluconeogenesis and starch and sucrose metabolism, in addition to geranyllinalool synthase (GES), linalool synthase (LIS) and benzyl alcohol O-benzoyltransferase (BAO), involved in terpenoid and benzenoid and benzoic acid biosynthesis. Based on the qPCR results, significant differences in expression levels were confirmed for *cpBGH3B*, *cpPFP*, *cpSUS*, *cpGES* and *cpLIS*. Except for *cpBGLU31*, these all presented higher expression levels in ‘RB1’ compared to ‘1B’. All results obtained from the qPCR analysis were consistent with the RNA-Seq data, except for *cpBGLU31*, in agreement with expression profiles inferred from transcriptome analysis.

## 3. Discussion

Commercial papaya varieties grown in Australia are relatively few; however, they do vary greatly in taste and aroma [3,11]. A previous profiling study highlighted the existence of undesirable tastes in current papaya varieties, resulting in the exclusion of a portion of the population as potential consumers [3]. To prioritise preferred flavours as the major breeding goal, we undertook a comprehensive exploration of the genetic mechanisms and biosynthetic pathways underlying these desired flavours. To identify a set of genes associated with sensory profiles, we conducted whole RNA sequencing and de novo transcriptome assembly on the distinctly flavoured papaya varieties ‘RB1’ (known for its sweet flavour and floral aroma) and ‘1B’ (characterised by a bitter taste and musty aroma) at both ripe and unripe stages [3]. The mean alignment rate of reads was better when mapped to the de novo transcriptome assembly when compared to the new updated genome, indicating that the tissue-specific de novo transcriptome improved the alignment rates. A previous study conducted by Wei and Wing in 2008 [12] annotated a total of 24,746 genes from the papaya genome, sequencing through the whole-genome shotgun method, and that figure is close to the number of genes annotated in our study (26,089 genes). To explore expression differences of specific genes further, more diverse tissue types (foliage or fruit skin) could be assessed for tissue-specific effects and potential allelic differences.

Except for one outlier of the ‘RB1-ripe’ sample, all samples were clustered with their corresponding replicates, demonstrating the assay’s repeatability. Also, these samples were well separated between varieties from the unripe stage to the ripe stage, indicating the different expression patterns among varieties during fruit development. Significant enrichment of DETs was observed in the oxidoreductase activity, metal ion transport and fatty acid biosynthetic process gene ontologies. The combination of oxidoreductase enzymes and metal ion transporters is mainly involved in regulating plant growth [13]. Since ‘RB1’ has an elongate fruit shape with red flesh and ‘1B’ has a round fruit shape with yellow flesh, it is reasonable that these genotypes have significant differences in oxidoreductase activity and metal ion transport. Moreover, fatty acid metabolism is the main pathway for the synthesis of esters, which are considered the major contributors to the unique papaya flavours [14]. The significant enrichment of DETs in the fatty acid biosynthetic processes indicates that ‘RB1’ and ‘1B’ may have distinct aroma profiles, which corresponds to the differences previously observed in sensory profiles of ‘RB1’ and ‘1B’ [3]. Assessing KOs corresponding to nodes of the KEGG pathways [15] highlighted lipid metabolism, amino acid metabolism and carbohydrate metabolism as the most enriched pathways in the samples assessed.

To further investigate the genes encoding key flavour-related traits in papaya, 118 DETs were identified from the de novo assembly. Several encode for enzymes involved in starch and sucrose metabolism (Figure 5), including invertase and sucrose synthase (SUS), key enzymes in sugar accumulation during fruit ripening [4]. Sucrose synthase (K00695; EC:2.4.1.13; six DETs) catalyses a reversible reaction with a role in sucrose synthesis and cleavage, while invertase (K01193; EC:3.2.1.26; four DETs) catalyses the hydrolysis of sucrose to fructose and glucose [16]. Six of the DETs encoding these two enzymes were more highly expressed in ‘RB1’ than in ‘1B’. In addition, beta-glucosidase (K01188 and K05349; EC:3.2.1.21; nine DETs) is an important enzyme participating in the release of glucose and aromatic compounds from cellulose [17]. Previous research in papaya flavour profiling mentioned that sweetness (contributed by glucose, fructose and sucrose) and aroma (produced by terpenes and esters) are the major factors involved in the biosynthesis of papaya flavour [3,18]. Accordingly, high numbers of transcripts encoding the enzyme beta-glucosidase in the sucrose metabolism pathway were detected in this study [17]. In plants, beta-glucosidase plays an important role in secondary metabolism, which initiates the biosynthesis of over 2000 monoterpene alkaloids and releases glucose from polysaccharides [19,20]. Based on the DET analysis in this study, ‘1B’ had higher expression levels of transcripts encoding beta-glucosidase than ‘RB1’, indicating that ‘1B’ potentially has a higher aroma intensity and less sweetness than ‘RB1’. This aligns with the previous studies in flavour profiling of these two papaya varieties [3,17].

Terpenes play major roles in the biosynthesis of papaya aroma [14,18]. The terpene biosynthesis pathways contain multiple terpene synthases to produce the different classes of terpenoids, such as monoterpenoids (C10), sesquiterpenoids (C15), diterpenoids (C20) and triterpenoids (C30) [22]. Among these, monoterpenoids contribute greatly to the flavours and aromas of plants [23]. Previously, linalool, which is a typical monoterpenoid, was found to majorly contribute to papaya flavour [3]. Further essential investigations are required to fully explore the terpenoid biosynthesis pathways in papaya to potentially identify the entire suite of enzymes that produce these volatiles that contribute to preferred papaya flavours and aromas.

There are two biosynthetic pathways involved in terpene biosynthesis: the mevalonate (MVA) pathway (present in the cytosol and mitochondria) and the 1-deoxy-d-xylulose/2-*C*-methyl-d-erythritol-4-phosphate (DOXP/MEP) pathway (present in the chloroplasts) [22]. The difference between the two pathways is quite clear: the MVA pathway generally produces sesquiterpenes and triterpenes, while monoterpenoids and diterpenoids are biosynthesised exclusively via the DOXP/MEP pathway [24,25]. DETs coding for enzymes with functional roles at the very beginning of both pathways have been identified in this study (Figure 6). For example, Acetyl-CoA C-acetyltransferase (ACAT) (K00626; EC:2.3.1.9; two DETs) initiates the MVA pathway by converting the starting material, acetyl-CoA, to acetoacetyl-CoA. Higher expression levels of these two DETs were found in ‘RB1’ compared to ‘1B’. Meanwhile, 1-deoxy-d-xylulose-5-phosphate synthase (DXPS) (K01662; EC:2.2.1.7; three DETs) functions in the DOXP/MEP pathway to convert the starting materials pyruvate and glyceraldehyde-3-phosphate to 1-deoxy-d-xylulose 5-phosphate. Higher expression levels of DETs encoding this enzyme were found in ‘1B’ than in ‘RB1’, which indicates that it produces more linalool and other monoterpenes than ‘RB1’, corresponding to the previous research in sensory analysis on these two papaya varieties [3]. The gene *VvDXS*, which encodes the enzyme DXPS, is positively related to high concentrations of aroma compounds, including linalool, α-terpineol, nerol and geraniol acid in grape [26,27]. A novel polymorphism site, P1678 (A/G), was identified in the *VvDXS* gene as a functional marker [27]. Further exploration of the nucleic acid variations among papaya homologs is warranted to develop associated function molecular markers.

To explore the expression patterns of DETs between two papaya varieties, eight candidate genes were evaluated through qPCR. Significant differences in expression levels in *cpBGH3B*, *cpPFP*, *cpSUS*, *cpGES* and *cpLIS* were observed between ‘RB1’ and ‘1B’ at the ripe stage, and all of them showed higher expressions in ‘RB1’ in comparison to ‘1B’ papaya fruit. A summary of the corresponding pathways and potential functions of these five candidate genes is presented in Table 2. 

Among these five genes, *cpBGH3B*, *cpPFP* and *cpSUS* encode enzymes that are involved in carbohydrate metabolism. Of these, *cpBGH3B* is predicted to be a beta-glucosidase BoGH3B-like gene involved in the degradation of cellulose to glucose; however, the BoGH3B-like gene has not been enzymatically characterised yet [20]. Meanwhile, ‘RB1’ showed significantly higher expression in *cpBGH3B*, which suggested that more glucose is generated in ‘RB1’ during this enzyme’s activity. Previous research on papaya flavour pointed out that a higher glucose amount results in a less sweet taste in ripe fruit, while ‘RB1’ is sweeter and has lower glucose levels than ‘1B’ [3]. Although the gene expression level of *cpBGH3B* suggested the opposite result, the final glucose level in fruit should be contributed by multiple enzymes’ activities and not defined by single gene expressions. Therefore, it is essential to identify more gene expressions involved in glucose biosynthesis.

The gene *cpPFP* is predicted to be a diphosphate-dependent phosphofructokinase (EC:2.7.1.90), which catalyses reversible interconversion between fructose-6-phosphate and fructose-1,6-bisphosphate [28]. The activity of this enzyme is negatively correlated with the amount of soluble sugar in rice and sugarcane [28,29]. In addition, *cpSUS* is a sucrose synthase in papaya and plays a key role in sucrose biosynthesis. Increased activities of sucrose synthase were found to be associated with the increased sucrose concentrations in strawberry and peach [30]. Although all three of these genes are related to the concentrations of sugar contents in papaya fruit, no clear relationship between these enzymes and the sugar composition of ripe papaya has been proposed and validated. Previous research illustrated that the sucrose concentration will only increase until the activity of sucrose phosphate synthase (SPS) exceeds the activities of sucrose-degrading enzymes [30]. Thus, to further correlate the results from sequence expression analyses with biochemical evaluation, it is important to consider the combined activities of multiple enzymes. Moreover, *cpGES* and *cpLIS* are the two genes encoding volatile synthesis enzymes. *cpGES* is predicted to be a geranyllinalool synthase (EC:4.2.3.144) involved in diterpenoid biosynthesis. It is a precursor to the volatile compound 4,8,12-trimethyl-1,3,7,11-tridecatetraene (TMTT), which is a herbivore-induced volatile and has a floral odour [31,32]. *cpLIS* is predicted to be an S-linalool synthase (EC:4.2.3.25) that initiates monoterpenoid biosynthesis. It is the precursor of (3S)-linalool, which has a floral scent and was identified to be emitted by the rare Brewer’s clarkia flower [33]. It is interesting to find that both genes related to the production of floral scents were more highly expressed in ‘RB1’ than in ‘1B’, while sensory analysis identified a greater floral aroma in ‘1B’ [3].

## 4. Materials and Methods

### 4.1. Plant Materials

Two Australian commercial cultivars, red-fleshed ‘RB1’ and yellow-fleshed ‘1B’, were harvested at ripening stages 1 and 3, based on the previously developed harvest ripening index (Figure 7), from a commercial papaya plantation in Mareeba, Australia (17.0° S, 145.4° E), in June 2021. Three fruits of each variety at each fruit maturity stage were used as biological replicates. A total of 12 fruit samples were treated with Sportak (active ingredient: >25% Prochloraz) to control postharvest disease, stored at 27 °C for one day and then transported in a refrigerated truck to the laboratory at Griffith University, Nathan Campus, Brisbane. Fruit flesh was cut into 5 mm pieces and flash frozen in liquid N_2_, then stored at −80 °C until processed for RNA extraction.

### 4.2. RNA Extraction from ‘RB1’ and ‘1B’

RNA was extracted from papaya flesh using NucleoZOL (Machery-Nagel Inc., Allentown, PA, USA) from approximately 50 mg of fruit flesh per sample according to the manufacturer’s instructions. Total RNA quantity and quality were assessed by a spectrophotometer (Microplate Fluorometer, Thermo Fisher Scientific, Waltham, MA, USA), using light absorbance of RNA at wavelengths of 260 nm (OD_260_) and 280 nm (OD_280_).

### 4.3. Transcriptome Profiling of the ‘RB1’ and ‘1B’ Genomes

#### 4.3.1. De Novo Assembly of the Transcriptome

A total amount of 2 µg of high-quality RNA per sample of ‘RB1’ or ‘1B’ was sequenced using the Illumina NovaSeq (S2 flow cell) platform (Illumina, Inc., San Diego, CA, USA); then, the Illumina bcl2fastq 2.20.0.422 pipeline was used through bcl2fastq Conversion Software (v2.20) to generate the sequence data by the Australian Genome Research Facility (AGRF). Short-read sequences from RNA-Seq were processed on the Queensland Research and Innovation Services (QRIS) Awoonga high-performance computing (HPC) Cluster under a Linux command-line environment, following the methods specified by Best Practices for De Novo Transcriptome Assembly with Trinity [34]. The raw sequence data were then processed to remove adaptors, as well as low-quality bases/reads using bbduk (from BBMap v38.34) followed by Trimmomatic (v0.40) and FastQC (v0.11.9) [35]. Trinity (v2.11.0) was then applied to the clean reads to de novo assemble the full transcriptomes of ‘RB1’ and ‘1B’ [36]. Additional scripts provided with Trinity (v2.11.0) were used to gather transcriptome statistics and calculate the N50 of the 90% most highly expressed transcripts (Ex90N50). The alignment rates of the clean reads when mapped to the de novo transcriptome and the non-GMO ‘Sunset’ genome published by Yue et al. [8] were compared using Bowtie2 (Galaxy Version 2.5.0 + galaxy0). A BLAST homology search was used to determine the coverage of curated and annotated genes from the Swissprot database that were found in each assembly.

#### 4.3.2. Gene and Protein Annotations

Open reading frames (ORFs) were predicted from the assembled transcripts using TransDecoder (v5.5.0) and then annotated against the nucleotide and protein databases at the NCBI (https://www.ncbi.nlm.nih.gov (accessed on 24 June 2022)) using a sequence alignment similarity search (BLAST) [37]. To reduce computation time, the taxonomic groups of the search were limited to flowering plants, fungi, viruses and bacteria. The predicted proteins were further annotated for protein families, motifs and gene ontology using InterProScan (v5.30-69.0) against a collection of databases, including SMART, Pfam, PANTHER, CDD and SFLD [38].

### 4.4. Differential Expression (DE) Analysis

Transcript expression quantitation was performed by Salmon (v1.5.2) to generate a transcript counts matrix. Transcripts that were differentially expressed between ‘RB1’ and ‘1B’ at the ripe stage were analysed by edgeR (the Empirical analysis of Digital Gene Expression in R, version 4.0.2) software and the Bioconductor package DESeq2 [39,40]. The specified thresholds for determining significant differences in transcript expression were a false discovery rate (FDR) < 0.05 and a fold change of 2.82 (|log_2_FC| > 1.5) [41]. Sample-related biases were identified through variance-stabilising transformation of the raw counts of reads using principal component analysis (PCA) and plotted into a between-sample distance matrix. The differentially expressed transcripts between ‘RB1’ versus ‘1B’ at the ripe stage were then used for gene ontology (GO) enrichment analysis using the GO2term R package (version 3.0.4), with a *p*-value ≤ 0.05 set to determine significantly enriched terms [42]. Additionally, Kyoto Encyclopedia of Genes and Genomes (KEGG) enrichment analysis was performed with KOBAS (version 3.0, http://kobas.cbi.pku.edu.cn/ (accessed on 15 August 2022)). The list of annotated KO (KEGG Orthology) numbers was then submitted to KEGG Mapper (version 5.0, https://www.genome.jp/kegg/mapper/ (accessed on 15 August 2022)) to determine the putative associated functional pathways [15].

### 4.5. Validation of Gene Expression Using Reverse Transcription Quantitative Real-Time PCR (RT-qPCR)

For RT-qPCR analysis, 1 μg of total RNA of each sample extracted at the ripe stage was taken for gDNA elimination and reverse transcription using a PrimeScript™ RT reagent Kit with a gDNA Eraser (Perfect Real Time) (Takara Bio, Kusatsu, Shiga, Japan). cDNA products were diluted to 1:50 in RNAse-free water for gene expression analysis. In addition, 10 μL of the 50× diluted cDNA from each sample was added into a new tube to make a pooled cDNA sample.

A total of eight differentially expressed transcripts (DETs) were selected to validate the expression profiles of the genes of interest (Appendix A). Of these DETs, five were annotated to be involved in sugar metabolism pathways (*cpBGLU42*, *cpBGLU31*, *cpBGH3B*, *cpPFP* and *cpSUS*) and three were related to volatile biosynthesis (*cpGES*, *cpLIS* and *cpBAO*). Primers to amplify each target sequence were designed using Primer3web (version 4.0.0) with the following criteria: a melting temperature (T_m_) of 60–65 °C and a PCR amplicon size of 100–200 base pairs (bp), a primer length of 17–25 nucleotides, and a GC content of 45–55%. Furthermore, two reference genes, *EF2* and *GAPDH*, were selected from Zhu et al. (2012) [43] as inter-run calibrators (IRCs). The PCR amplification efficiency (E) of each primer pair was assessed using serially diluted pooled cDNA samples (10^0^, 10^−1^, 10^−2^, 10^−3^ and 10^−4^) and calculated through Bio-Rad CFX manager (version 3.0) software.

RT-qPCR was carried out to verify the expression patterns of candidate genes on a Bio-Rad CFX96 real-time PCR detection system (Bio-Rad laboratories). Each RT-qPCR reaction contained 12.5 μL of 2× SYBR Premix Ex TaqTM II (TIi RNaseH Plus, TaKaRa Bio, Japan), 0.4 μM of each primer and 2 μL of diluted cDNA template. The following cycle was used: an initial step of 95 °C for 30 s (denaturation), followed by 40 cycles of 95 °C for 5 s, 60 °C for 30 s (fluorescence reading), followed by a melt curve analysis at 65–95 °C every 0.5 °C for 10 s. All reactions were carried out in three technical replications.

### 4.6. RT-qPCR Data Analysis

Cq data were exported from the Bio-Rad CFX manager software (version 3.1), and the samples that did not amplify or that produced Cq values under 5 or over 40 cycles were removed. After that, gene expression data were imported into Factor-qPCR software version 2020.0 [44] and normalised to account for between-run variations. The Delta-Delta-Cq (ΔΔCq) algorithm [45] was applied on the normalised Cq values and the geometric mean of the selected reference genes to determine the differential expressions between ripe ‘RB1’ and ‘1B’ samples. A Mann–Whitney test at the 5% level of significance was performed using Microsoft Excel (Microsoft 365, version 2205) to compare the significant expression differences between ‘RB1’ and ‘1B’.

## 5. Conclusions

In this study, whole RNA sequencing followed by de novo transcriptome assembly and differential expression analysis identified distinct and shared sequences associated with flavour in two Australian commercial papaya varieties, ‘RB1’ and ‘1B’. A total of 180,368 transcripts were generated and annotated to 26,089 genes. The analysis of transcriptome sequencing revealed 118 DETs involved in the pathways of carbohydrate metabolism and fatty acid and terpenoid biosynthesis metabolism. Key genes, including *cpBGH3B*, *cpPFP*, *cpSUS*, *cpGES* and *cpLIS*, were more highly expressed in ‘RB1’ than in ‘1B’, resulting in higher sugar accumulation and terpenoid generation in ‘RB1’. This may lead to ‘RB1’ being the preferred tasting variety with a sweet taste and a pleasant aroma, which aligns with the flavour profiling of papaya in a previous study. Additionally, qPCR validated the expression profiles at the ripe stage and highlighted these as potential molecular targets to develop markers for future selective breeding of papaya varieties with preferred flavours.

## Figures and Tables

**Figure 1 ijms-25-03046-f001:**
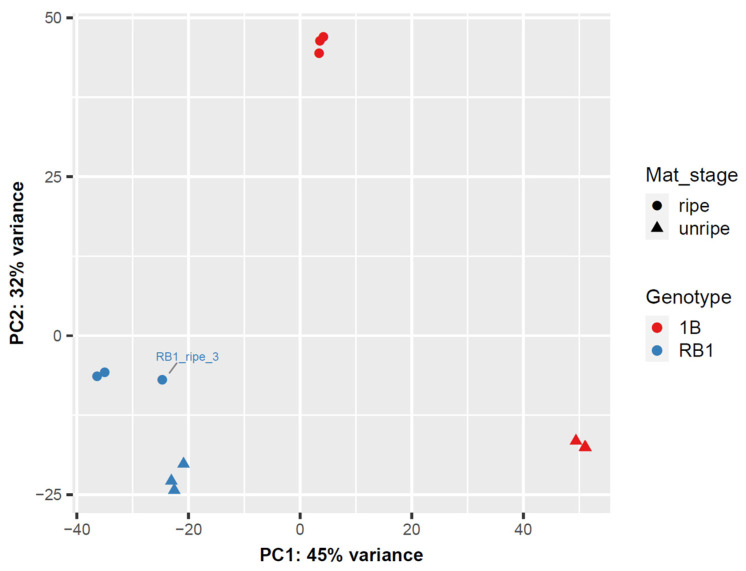
Principal component analysis of the blind variance-stabilised counts. Samples were categorised by genotype (indicated by marker colours; red for 1B and blue for RB1) and maturity stages (indicated by marker shapes; round for ripe and triangle for unripe stages).

**Figure 2 ijms-25-03046-f002:**
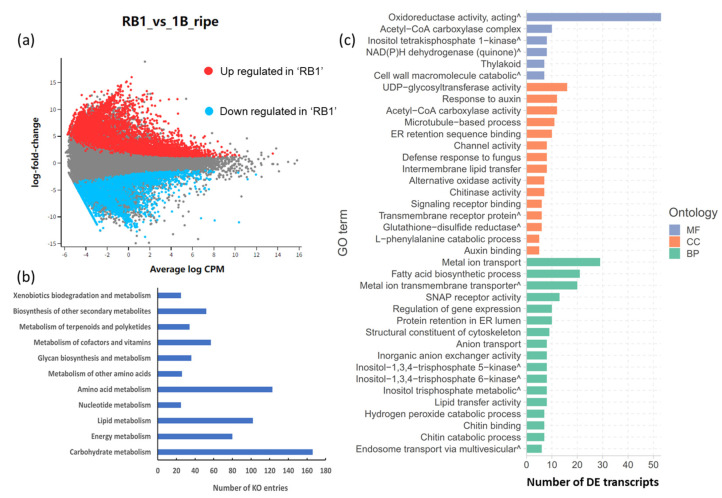
Analysis of DETs between ‘RB1’ and ‘1B’. (**a**) Volcano plot of DETs. (**b**) Classification of the KEGG Ontologies sub-categories involved in papaya metabolism. (**c**) Gene Ontology (GO) enrichment analysis based on DETs expressed between ‘RB1’ and ‘1B’ at ripe stage. BP: biological processes, CC: cell cycle, MF: molecular function (^ indicates that the full term name was shortened; see full names in Appendix A).

**Figure 3 ijms-25-03046-f003:**
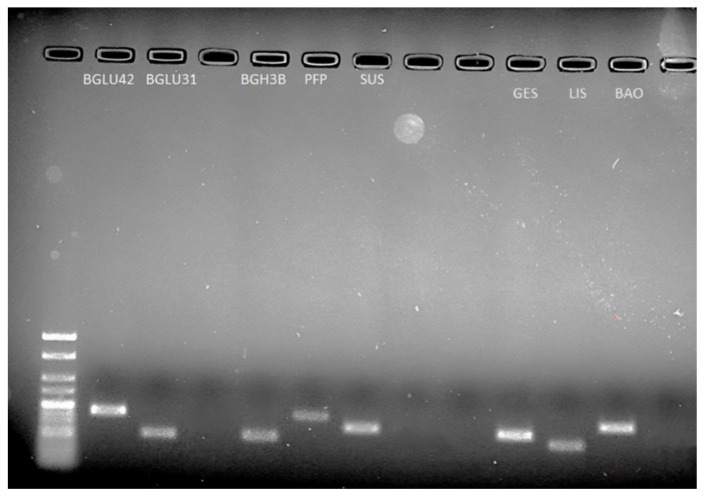
Validation of eight candidate genes in the pooled cDNA of RB1 and 1B by electrophoresis.

**Figure 4 ijms-25-03046-f004:**
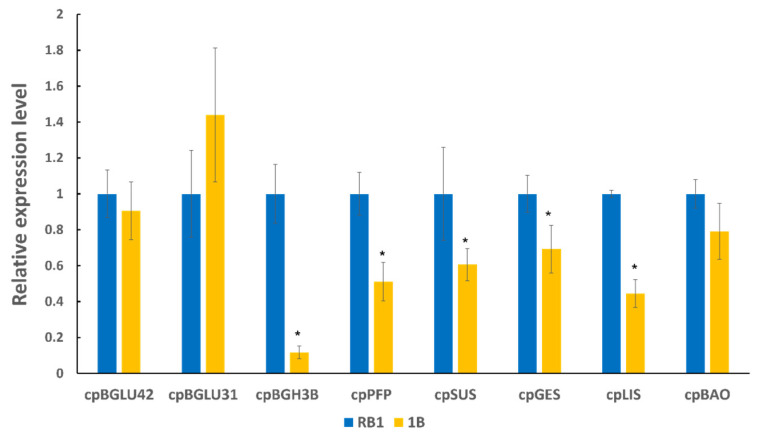
Expression profiles of eight candidate genes determined by qRT-PCR (* indicates significant differences in expression levels between 1B and RB1 based on a pairwise *t*-test at *p* < 0.05).

**Figure 5 ijms-25-03046-f005:**
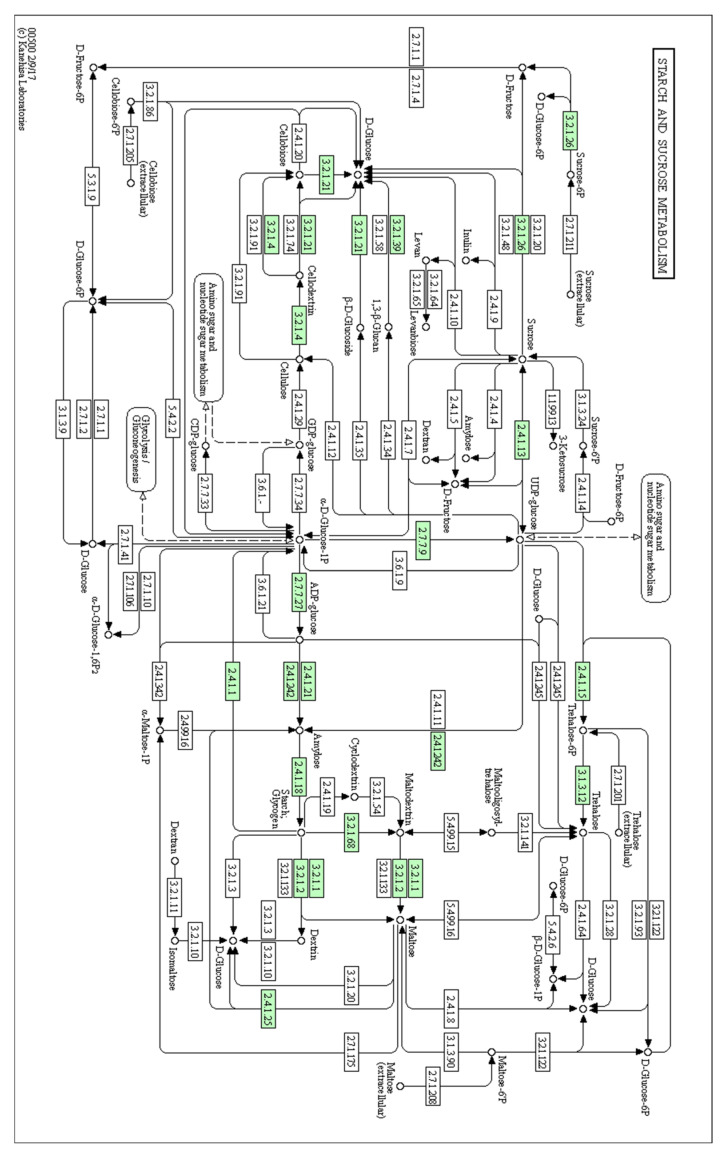
Starch and sucrose metabolism pathway. The papaya DETs encoding enzymes are highlighted in green blocks [21].

**Figure 6 ijms-25-03046-f006:**
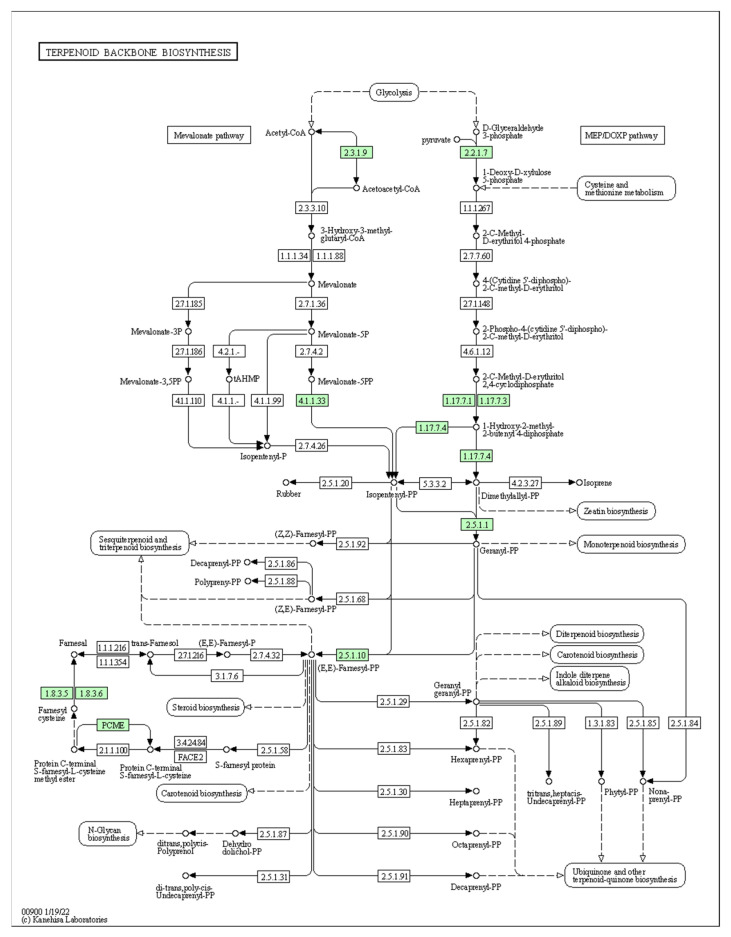
Terpenoid backbone biosynthesis. The papaya DETs encoding enzymes are highlighted in green bricks [21].

**Figure 7 ijms-25-03046-f007:**
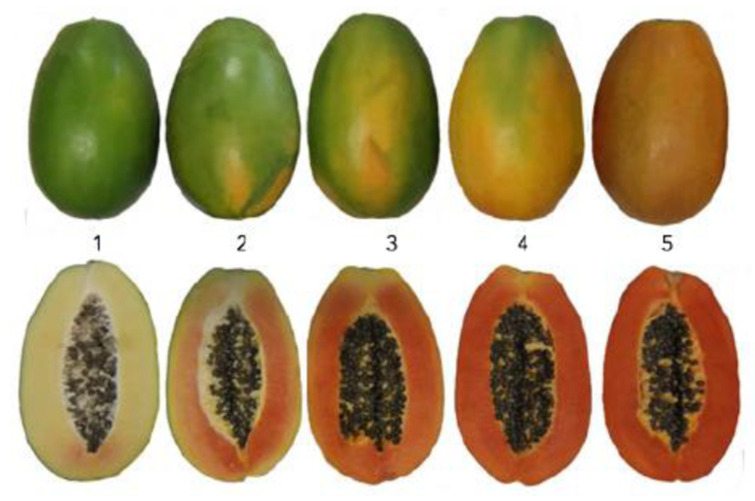
Five fruit maturity and ripening stages [11] (1 = mature green, 2 = 25% colour, 3 = 50% colour, 4 = 75% colour, 5 = fully ripe).

**Table 1 ijms-25-03046-t001:** Transcriptome assembly details of papaya ‘RB1’ and ‘1B’.

Feature	Value
Mean sequence quality (phred score)	36
Total assembled bases (n)	3.67 × 10^8^
Total assembled contigs (n)	180,368
Average contig length (bp)	2033.27
Median contig length (bp)	1345
Longest contig length (bp)	26,587
Contig N50 (bp)	3555
Contig N75 (bp)	2083
Contig Ex90N50 (bp)	2450
Assembly GC content (%)	47.04
Mean alignment rate to de novo assembled genome	98.18%
Mean alignment rate to the non-GMO ‘sunset’ genome	81.17%

**Table 2 ijms-25-03046-t002:** A summary of the five significantly differentially expressed candidate genes.

Candidate Genes	Corresponding Pathway	Potential Functions
*cpBGH3B*	Starch and sucrose metabolism pathway	Involved in the degradation of cellulose to glucose
*cpPFP*	Fructose and mannose metabolism	Catalyses reversible interconversion between fructose-6-phosphate and fructose-1,6-bisphosphate
*cpSUS*	Starch and sucrose metabolism pathway	Catalyses the reversible cleavage of sucrose into fructose
*cpGES*	Diterpenoid biosynthesis pathway	A precursor to the volatile compound TMTT, which has a floral odour
*cpLIS*	Monoterpenoid biosynthesis pathway	A precursor of (3S)-linalool, which has a floral scent

## Data Availability

Assembled RNA-Seq data have been deposited in the Gene Expression Omnibus (GEO) database with the accession code GSE213436.

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
