# Peer review of "Identification and Validation of Key Genes Related to Preferred Flavour Profiles in Australian Commercial Papaya (Carica papaya L.)"

_ijms, 2024, doi:10.3390/ijms25053046_

Round 1
Reviewer 1 Report
Comments and Suggestions for Authors
The manuscript by Zhou et al. identified and validated some important genes related to preferred flavour profiles in Australian commercial papaya (Carica papaya L.). I have enjoyed reading the MS. This paper would be useful for papaya improvement program, However I have some major comments for the revision.
1. What is preferred flavor? And how can author say they identified genes specifically related to preferred flavor?
2. Line 50, set of genes -> suit of genes.
3. Line 17, what is extreme flavored papaya? Please correct this term throughout the MS
4. Author should perform statistical test for qRT-PCR experiment to show the significant difference among two samples in the study.
5. The quality of figure 4 is poor, should upload high pixel image.
6. Figure 6, should have scale bar.
7. Figure 6, What are the numbers (1,2,3..) for? What they signify? No mention in legends.
8. On what basis eight DETs where selected for qRT PCR validation?
Comments on the Quality of English LanguageNeed to improve the writings to make simpler sentences.
Reviewer 2 Report
Comments and Suggestions for Authors
I would ask the authors how they planned this study and how it differs from their previous study Int. J. Mol. Sci. 2022, 23(11), 6313; https://doi.org/10.3390/ijms23116313
I fail to understand why the authors have not done GCMS to estimate the volatile command before proceeding with such a study
Before proceeding to sequencing, the RIN value of the sample should be given for quality check and it will also precise the results
Authors should give a clear phred quality score of each sample in table 1. The GC content of each sample was very low as indicated table , justify?
the result should also include a clear gel picture of q PCR validation of genes it will enhance the reader's understanding ability.
How this study does not address all the concert solution until it should have been planned properly as the authors have not done the preliminary study for volatile commands, further study is on flavor profile however author fails to address the all pathways involved in it without proper planning of crop stage and sample with replicates. only two stages may not be sufficient to address this, hence recommended to take more stages with different plants rather than taking a single plant's fruits.
we expect mapman analysis of the DET distribution which also missing
study should end with develop,ent of genic based markers for future use is missing?
considering all the above points this study is not complete in this ms at this stage?
Comments on the Quality of English Language
I would ask the authors how they planned this study and how it differs from their previous study Int. J. Mol. Sci. 2022, 23(11), 6313; https://doi.org/10.3390/ijms23116313
I fail to understand why the authors have not done GCMS to estimate the volatile command before proceeding with such a study
Before proceeding to sequencing, the RIN value of the sample should be given for quality check and it will also precise the results
Authors should give a clear phred quality score of each sample in table 1. The GC content of each sample was very low as indicated table , justify?
the result should also include a clear gel picture of q PCR validation of genes it will enhance the reader's understanding ability.
How this study does not address all the concert solution until it should have been planned properly as the authors have not done the preliminary study for volatile commands, further study is on flavor profile however author fails to address the all pathways involved in it without proper planning of crop stage and sample with replicates. only two stages may not be sufficient to address this, hence recommended to take more stages with different plants rather than taking a single plant's fruits.
we expect mapman analysis of the DET distribution which also missing
study should end with develop,ent of genic based markers for future use is missing?
considering all the above points this study is not complete in this ms at this stage?
Round 2
Reviewer 2 Report
Comments and Suggestions for Authors
this study was designed to uncover genes related to the consumer preferred flavours, which could be the major metabolic switches in sucrose metabolism and aromatic compounds production pathways and will be further investigated for use in future selective breeding strategies through assessment in broader germplasm and growth environments. Considering the comments addressed by the author and future prespective , it may be considered for the publication. please add the Gel pic of qPCR validation of genes on the pooled cDNA of RB1 and 1B in the main text for clear understanding to readers.
Author Response
Thank you very much for your suggestions. The gel pic has been added to the main text as Figure 3 (line 155).